



# The Role of Weather Regimes for Subseasonal Forecast Skill of Cold-Wave Days in Central Europe

Selina M. Kiefer[1], Patrick Ludwig[1], Sebastian Lerch[2], Peter Knippertz[1], and Joaquim G. Pinto[1]

[1]Karlsruhe Institute of Technology (KIT), Institute of Meteorology and Climate Research Troposphere Research (IMKTRO), Kaiserstraße 12, 76131 Karlsruhe
[2]Karlsruhe Institute of Technology (KIT), Institute of Statistics (STAT), Kaiserstraße 12, 76131 Karlsruhe

**Correspondence:** Selina M. Kiefer (selina.kiefer@kit.edu)

**Abstract.** Weather regimes (WRs) represent the large-scale tropospheric flow and therefore may contain useful information about the subseasonal predictability of cold waves, one of the most severe weather extremes in Central Europe. Firstly, we investigate in how far the succession of WRs during a forecast can be used to explain skill differences of forecasts initialized during different WRs. As an example, we use the skill differences of mean-bias-corrected 14-day reforecasts of the European Centre for Medium-Range Weather Forecasts for the occurrence of wintertime cold-wave days in Central Europe. Reforecasts initialized during the WR Greenland Blocking (GL; characterized by a high pressure system over Greenland) show the best Brier skill while those initialized during the WR Scandinavian Trough (ScTr; characterized by a low pressure system over Scandinavia) show the worst skill compared to a climatological ensemble for the winters 2000/2001-2019/2020. We find, that for forecasts initialized during GL, more often WR succession which follow typical climatological pattern are found during the 14 days of forecasts than for forecasts initialized during ScTr. We suggest that this is one of the main reasons for an increased forecast skill of predictions initialized during GL in contrast to predictions initialized during ScTr. Secondly, we analyze the WR succession for the best (worst) predicted days within the observed cold waves in the winters 2000/2001-2019/2020 independent from the WR present at initialization. We find, that forecast skill is significantly higher, when the European Blocking WR (characterized by a high pressure system over the British Isles and southern Scandinavia) is present a few days before the predicted cold-wave day. These results can be used to assess the reliability of cold-wave day predictions at the subseasonal lead time of 14 days.

## 1 Introduction

Cold waves are among the most important and severe wintertime weather extremes affecting Central Europe. For example, 600 fatalities were reported across Europe due to extreme cold temperatures and heavy snowfall during the cold wave occurring in February 2012 (DWD, 2012). Furthermore, transportation and energy supply were severely disrupted during this period. To minimize the negative impacts associated with cold waves, it is important to take timely measures. Therefore, an accurate prediction of the occurrence of cold waves in advance is highly desirable. Unfortunately, numerical weather prediction (NWP) models such as the subseasonal-to-seasonal (S2S) forecast model from the European Centre for Medium-Range Weather Forecasts (ECMWF) generally have very limited skill on that timescale, which comprises two to four weeks in advance (White




et al., 2017). This is due to the decreasing importance of initial conditions and only slowly increasing importance of boundary conditions of forecasts.

Recently, machine learning (ML) models for postprocessing NWP forecasts have become a method of interest on these timescales. Horat and Lerch (2024), for example, use convolutional neural networks (CNNs) for the correction of systematic errors of global ECMWF predictions for bi-weekly aggregations of precipitation and temperature. Scheuerer et al. (2020)

use artificial neural networks and CNNs for postprocessing subseasonal predictions of ECMWF's reforecast ensemble of precipitation accumulated over California. Kiefer et al. (2024) demonstrate the ability of probabilistic Random Forest (RF)-based postprocessing models to generate improved forecasts of Central European mean wintertime 2-meter temperatures and the occurrence of cold-wave days, using ECMWF's reforecast ensembles and reanalysis data as input.

In contrast to single meteorological variables, large-scale atmospheric patterns such as the North Atlantic Oscillation (NAO)

can be predicted skillfully beyond a lead time of 10 days without the need for postprocessing (Ferranti et al., 2018). The predicted NAO pattern can then be used as an indicator of forthcoming surface weather. The NAO is characterized by two centers of action, the Icelandic Low and the Azores High (Pinto and Raible, 2012). Depending on the strength of these pressure systems, the strength of the westerly winds over the eastern North Atlantic Ocean varies (Pinto and Raible, 2012) and the large-scale atmospheric flow is either more zonal (NAO+) or more meridional (NAO-), and thus determines the surface

weather conditions over Europe (Benedict et al., 2004). Large-scale atmospheric patterns like the NAO are persistent and quasi-stationary for several days (Hannachi et al., 2017), and can even persist longer during so-called "windows of opportunity", during which teleconnections tied to large-scale feedbacks in the climate system are present (Mariotti et al., 2020). One example, where predictability is enhanced into the subseasonal timescale is described in Kautz et al. (2020). They show that the stratospheric polar vortex influences the state and persistence of the NAO during the late winter of 2018. When the

stratospheric polar vortex weakens, and in some cases even changes its wind direction from westerly to easterly during a sudden stratospheric warming (SSW) event, the developing flow anomalies can propagate downward into the troposphere, where they influence the large-scale circulation and subsequently surface weather patterns (Baldwin et al., 2003). Often, the large scale-circulation is resembling a so-called Greenland Blocking (GL) or Atlantic Trough (AT) pattern in the weeks after the SSW event (Domeisen et al., 2020). The GL regime is characterized by a strong quasi-stationary high pressure system located

over Greenland. During this situation, cold airmasses originating from the Arctic or Siberia are transported into Northern and Central Europe, frequently leading to cold waves. In contrast, the AT regime is characterized by a strong low pressure system west of the British Isles. During this situation, warm airmasses are transported zonally across the Atlantic Ocean leading to mild temperatures in Central Europe. Both regimes can be characterized using the weather regime (WR) classification for the Euro-Atlantic region proposed by Grams et al. (2017). It consists of seven distinct weather regimes and a so-called "No" regime

class, which collects large-scale tropospheric flow patterns close to climatology.

Four of the WRs are characterized by blocking, which means that a high pressure system is quasi-stationary and persistent for more than five consecutive days, usually leading to cool temperatures over Central Europe during winter. The other three regimes are characterized by a stronger than usual zonal flow, transporting warm temperatures towards Central Europe.





In a previous study (Kiefer et al., 2024), we demonstrate for the mean-bias-corrected ECWMF's S2S reforecasts and RF-
based postprocessing models that the skill of subseasonal forecasts of the occurrence of wintertime cold-wave days in Central
Europe is dependent on the WR present at initialization. In this study, we analyze in how far the succession of WRs during the
forecast, evaluated for the winters 2000/2001-2019/2020, can explain these differences in skill. Besides the analysis of both,
the occurrence of (non-) cold-wave days, we furthermore focus on the differences between the best and worst predicted days
within cold waves. We aim to answer the following research questions:

1. In how far can the WR succession during a forecast be linked to subseasonal forecast skill?

        2. What are the differences in the WR succession before the best (worst) predicted days within cold waves?

In order to answer the first question, we start by illustrating at the example of the winters 2010/2011 and 2017/2018 which
parameters can be used to describe WR successions. These winters contain multiple episodes of the WRs GL and ScTr. When
present at initialization time, GL is associated with the best, ScTr with the worst forecast skill of ECMWF's S2S reforecasts at
a lead time of 14 days compared to a climatological benchmark ensemble during the winters 2000/2001-2019/2020. Since the
differences in skill of the named subsets of forecasts (GL or ScTr at initialization) are significant, we use these to demonstrate
in how far the WR succession during a forecast can be linked to ther skill.

To answer question two, we focus on the 45 cold waves observed during the winters 2000/2001-2019/2020. Thereby, we
analyze the best and worst predicted cold-wave days of the mean-bias-corrected ECMWF's S2S reforecasts and a representative
RF-based postprocessing model independent of the WR at initialization.

Section 2 of this paper introduces the data and Sect. 3 the methods that have been used. The results are described in Sect. 4.
Section 6 provides a discussion, conclusions and an outlook.

## 2  Data

### 2.1  Climatological benchmark ensemble

A climatological benchmark ensemble is constructed based on E-OBS V23.1e observational data (Cornes et al., 2018). To that
end, the daily mean 2-meter temperature ($tg$) over land, which is interpolated from station data to a regular latitude-longitude
grid, is retrieved with a resolution of $0.1 \times 0.1°$ for the months November to April from the winters 1970/1971 until 1999/2000.
These winters are preceding the evaluation period used in this study which consists of the winters 2000/2001-2019/2020. To
construct the climatological ensemble predictions, the temperature time series of each winter is used as one ensemble member,
resulting in a 30-member ensemble. Each ensemble member is spatially averaged over the region between 3°E to 20°E and
45°N to 60°N, whereby the first coastal grid points and mountainous regions with altitudes above 800 m are excluded to
create a more homogeneous area (Fig. 1). In the following, this region is referred to as "Central Europe". To take the temporal
uncertainty of subseasonal predictions into consideration, the climatological ensemble prediction is smoothed temporally by a
7-day running mean. Afterwards, every day is classified as a cold-wave or non-cold-wave day by using the definition by Smid





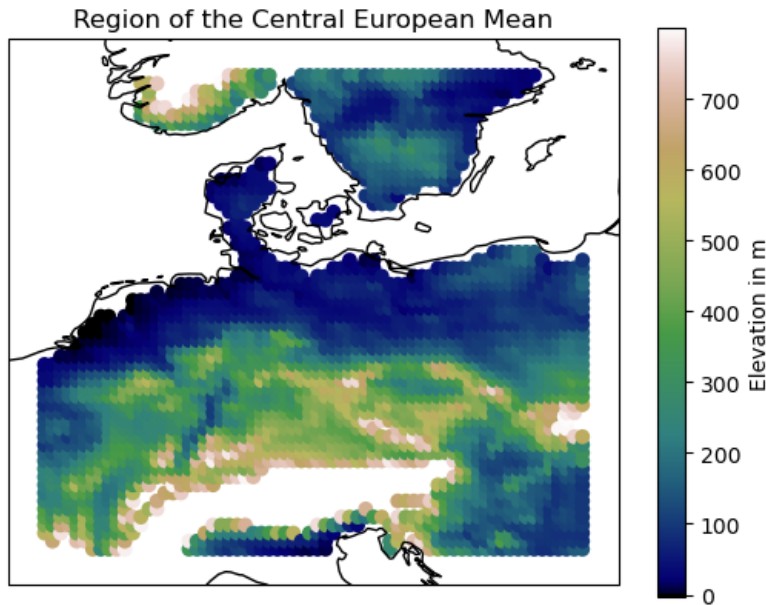

**Figure 1. Visualization of the region of Central Europe as used for averaging in this study.** The map is created using the python package "Cartopy" (Met Office, 2010 - 2015).

et al. (2019), based on the 10th percentile of climatological mean temperatures, with two slight adaptions. Instead of calculating the thresholds based on the 2-meter daily minimum temperatures, we use the daily mean 2-meter temperatures. Furthermore, we use the winters 1970/1971 until 1999/2000 as the reference period instead of the years 1981-2010 since the latter overlaps with the evaluation period used in this study. A cold wave is defined as at least three consecutive days with temperatures below the calculated threshold. We define a day as a cold-wave day when it is part of a cold wave. A detailed reasoning of the
modifications to the original approach by Smid et al. (2019) and the used thresholds can be found in Kiefer et al. (2023).

## 2.2   ECMWF's S2S reforecasts

ECMWF's S2S 2-meter temperature reforecasts (Vitart et al., 2017) are retrieved for the area of Central Europe for the month November to April of the winters 2000/2001 until 2019/2020 at a lead time of 14 days. The reforecasts are initialized every Monday and Thursday at 00 UTC and predict the daily mean temperature at a spatial resolution of $1.5 \times 1.5°$ latitude-longitude.
For further use, the reforecasts are spatially averaged over Central Europe and the occurrence of cold-wave days is calculated. In this study, we use the term "target date", which is equivalent to the valid date of the reforecasts.

## 2.3   Weather regime data

The WRs are identified based on the $500\,\mathrm{hPa}$ geopotential height from ERA5 reanalysis data (Hersbach et al., 2020a). As proposed in Grams et al. (2017), the WRs are calculated over the Euro-Atlantic region, 80°W to 40°E and 30°N to 90°N,



using Empirical Orthogonal Functions (EOFs) and k-means clustering of the first seven EOFs. This results in seven categorical WRs. To identify to which extent the current atmospheric large-scale flow is consistent with the WRs, the continuous WR index can be used. This index is constructed by projecting the current atmospheric flow situation onto the cluster mean of the seven WRs. Whenever this index is above 1, a WR is considered to be present. Thereby, the current atmospheric large-scale flow can have characteristics of multiple WRs at the same time. The WR with the highest positive WR index is the dominating regime. A WR is chosen as the categorical WR of a timestep if its WR index value is above 1 for more than five consecutive days and at least once during this time the dominating WR. In case none of the WRs fulfills these criteria at a timestep, the categorical WR of that timestep is called the "No" regime. In the data used in this study, which comprises the time period between 11 January 1979 and 13 August 2022 at a three-hourly resolution, 30% of the cases belong to this category. The other seven regimes are divided into the so-called blocked and cyclonic regimes. As shown in Fig. 1 in Büeler et al. (2021), the blocked regimes are characterized by a strong quasi-stationary high pressure system, which leads to a meridional airflow and thus to cool temperatures in Central Europe. In case of the European Blocking (EuBL) regime, which occurs on roughly 10% of the days, a high pressure system is located over the British Isles and Southern Scandinavia. In case of the Greenland Blocking (GL) regime, high pressure is located over the south of Greenland and the Labrador Sea. This situation is also found on approximately 10% of the days. During the Scandinavian Blocking (ScBL) regime, which occurs in circa 11% of the cases, a high pressure system is located over northern Scandinavia and during the Atlantic Ridge (AR) regime, which occurs on roughly 9% of the days, a high pressure system is located south of Iceland. The cyclonic regimes, related to low pressure systems, are characterized by a zonal flow of milder and wetter air across the North Atlantic Ocean.

In case of the Atlantic Trough (AT) regime, which is present in approximately 9% of the days, a low pressure system is located west of the British Isles. During the Scandinavian Trough (ScTr) regime, a low pressure system is located between Iceland and Scandinavia. This situation occurs in roughly 11% of the days. During the Zonal (ZO) regime, which is present on approximately 9% of the days, a low pressure system is located between the south of Greenland and Iceland.

## 3 Methods

### 3.1 Mean-bias-correction of ECMWF's S2S reforecasts

In order to correct for biases in ECMWF's S2S 2-meter temperature reforecasts, a bias correction based on the E-OBS observational data is used. Therefore, the daily mean of the reforecast ensemble is calculated. From these daily means, the observed temperature over Central Europe from the E-OBS dataset is subtracted, resulting in a time series of biases. Then, ECMWF's S2S reforecast ensemble is bias-corrected by applying a leave-one-winter-out approach. This means, that the time series of biases is temporally averaged over all days of 19 out of the 20 winters between 2000 and 2020. Then, the calculated mean is subtracted separately from every ensemble member of ECMWF's S2S reforecast of the left out winter. As an example, the winter 2010/2011 is bias-corrected by the temporal mean of the time series of biases of the winters 2000/2001 until 2009/2010 and 2011/2012 until 2019/2020.



### 3.2 Random Forest Classifier-based postprocessing model

A representative Random Forest Classifier- (RFC) based postprocessing model from Kiefer et al. (2024) is used in this study
to predict the binary occurrence of (non-) cold-wave days over Central Europe in winter. This model takes the ECMWF's S2S

140 reforecast ensemble (Vitart et al., 2017) and ERA5 reanalysis data (Hersbach et al., 2020a, b) as input. The predictors are
chosen based on meteorological domain knowledge and include the zonal wind in 10 and 300 hPa, the geopotential in 100 ,
250 (ERA5) or 300 (ECMWF's S2S reforecasts), 500 and 850 hPa height, the temperature and specific humidity in 850 hPa
height as well as the mean sea level pressure. To reduce computational costs, the predictors are retrieved only for the area
between 60°W to 60°E and 20°N to 80°N, which comprises the areas of the North Atlantic Ocean and parts of Eurasia, which

are known to affect Central European winter weather. A more detailed reasoning of the predictor selection can be found in
Kiefer et al. (2023).

   ERA5 reanalysis data is retrieved at 00, 06, 12 and 18 UTC for the months of October to April 2000 to 2020. The data are
daily averaged and have a resolution of $1.5° \times 1.5°$ latitude-longitude. ECMWF's S2S reforecasts have the same resolution but
are retrieved directly at 00 UTC. All meteorological fields are preprocessed before model training by computing the minimum,

mean, maximum and variance of each field. In case of ECMWF's S2S reforecasts, only the ensemble information (minimum,
mean and maximum and their variances) instead of each individual ensemble members is taken into account. Furthermore, the
minimum, mean, maximum and variance of the 2-meter temperature reforecast ensemble, averaged over Central Europe, is
added as a predictor. The month is also added in order to account for the seasonality of temperatures and thus the occurrence
of cold-wave days in winter. As the ground truth used for verification, binary occurrences of cold-wave days over Central

European in the E-OBS dataset (Cornes et al., 2018) for the months November to April 2000-2020 are used. Training is done
using a leave-one-(winter-)out cross-validation approach. This means that for every of the 20 winters between 2000-2020, a
separate model is trained. For example, the model predicting the occurrence of cold-wave days of the winter 2015/2016 is
trained on the combined data of the winters 2000/2001 to 2014/2015 and 2016/2017 to 2019/2020. The model setup is based
on the Python package "skranger" (Flynn, 2020-2021) with the change of having 1000 decision trees (instead of 100) and a

minimal node size of 5 (instead of 1). Further information about the model setup can be found in Kiefer et al. (2023). We use
the described RFC-based postprocessing model in this study, since it yields the best predictions of the binary occurrence of
cold-wave days in the 20-winter mean of the winters 2000/2001 to 2019/2020 at a lead time of 14 days compared to the other
RFC-based postprocessing models used in Kiefer et al. (2024).

### 3.3 Brier Score

The Brier Score (BS) is used to evaluate the predictions of the climatological benchmark ensemble, the mean-bias-corrected
ECMWF's S2S reforecasts and the RFC-based postprocessing model with respect to the E-OBS observational Central Euro-
pean daily mean 2-meter temperature. It can be written as (Brier, 1950)

$$BS_{\mathrm{F}}(y) = \left(F(1) - \mathbf{1}\{y = 1\}\right)^2, \tag{1}$$



where $F(1) = \hat{P}(y = 1)$ is the forecast probability of occurrence of a cold-wave day. The lower the BS value, the better the forecasts. A perfect prediction has a BS value of zero.

We use the daily BS difference of the climatological benchmark ensemble and the mean-bias-corrected ECMWF's S2S reforecasts as a daily measure of skill:

$$BS_{\text{diff}} = BS_{\text{benchmark}} - BS_{\text{model}}. \tag{2}$$

Since a lower BS denotes better forecasts, a positive BS difference indicates better predictions of the ECMWF's S2S reforecast ensemble, a negative one a better prediction of the benchmark.

We estimate the significance of differences between groups of predictions applying a Welch's t-test which we computed using the Python package "scipy" (The SciPy community, 2008-2024). If the calculated p-value is smaller than 0.05, we assume the differences between the groups to be significant.

## 4 Results

In Kiefer et al. (2024), we compare the skill of subseasonal forecasts of the occurrence of cold-wave days in Central Europe depending on the WR present at initialization. Among other results, we find that the mean-bias corrected ECMWF's S2S reforecasts with a lead time of 14 days show a significantly better skill in the 20-winter mean when initialized during the GL regime in comparison to the ScTr regime. We chose the example of the mean-bias-corrected ECMWF's S2S reforecasts since these predictions rely on the physics of the atmosphere which are also represented by the WRs (Subsect. 4.1). Furthermore, we investigate potential differences in the WR successions before the best (worst) predicted days within cold waves independent of the WR present at initialization (Subsect. 4.2).

### 4.1 Linkage of WR successions during the forecast to subseasonal forecast skill

In order to increase the sample size, we use in the following all days of the winters 2000/2001-2019/2020 instead of only the days where ECMWF's S2S reforecasts are initialized. This leads to a mixture of "hypothetical" forecasts (since no reforecasts are initialized at that date) and "real" forecasts. For better reading, we use only the term "forecasts" in the following. We assume that the number of days on which each regime is present, 757 in case of GL and 713 in case of ScTr during winters 2000/2001-2019/2020, are similar enough to make a fair comparison.

### 4.1.1 WR successions during two illustrative winters

To illustrate how the skill evolution of subseasonal forecasts might depend on the WR present at initialization, we perform two case studies. Since we focus on forecasts initialized during the GL and ScTr regime, we select two winters which contain both regimes. These are, among others, the winters of 2010/2011 and 2017/2018.

In the former, GL is the dominant regime at initialization for forecasts predicting the occurrence of (non-)cold-wave days between mid-November and mid-January (Fig. 2 (a)). Except for a single day, the BS difference is always positive during that





time, indicating that forecasts of the mean-bias -corrected ECMWF's S2S reforecasts initialized during the GL regime are
more skillful than the climatological benchmark during this time, which features many cold-wave days. The ScTr regime is
present at initialization in the beginning of November and mid-March. During these periods, the BS difference is either slightly
positive or negative.

In case of the winter 2017/2018, the ScTr regime is present at initialization five times, roughly once every month (Fig. 2
(b)). Except for forecasts with target dates in the beginning of February, where the BS difference is slightly positive, the BS
difference is either close to zero or slightly negative. The GL regime is present three times at initialization during this winter,
once for forecasts predicting the occurrence of (non-)cold-wave days in November and twice in February/March. Except for
the last one, the BS difference is positive or close to zero.

A striking difference between the GL regimes and the ScTr regimes in the analyzed two winters is their duration. The
GL regimes tend to be more persistent such that the number of regime successions during the lead time of the forecast is
smaller, which is a possible explanation for the skill differences. For computational simplicity, we measure this as the number
of transitions between regimes (parameter one). During the GL regime, the WR index has higher values than during the ScTr
regimes and there is a tendency for the other regimes to show more negative values of the WR index during the GL regimes
than during the ScTr regime. This means that the number of active WRs at each day of lead time is less for the GL regime,
a second possible explanation for the skill differences. We measure this as the overall number of active WRs at each day of
lead time during all forecasts of the winters 2000/2001-2019/2020 (parameter two). For simplicity, we only take the most
prominent WRs into consideration as represented by the categorical WR of the respective day. The third difference between
the two regimes at initialization is the subsequent regime that occurs. In case of GL, either the AR or AT regime is following.
In case of ScTr, the subsequent regime is either the GL, EuBL, AT or ZO. Thus, the third considered parameter is the actual
succession of WRs during the forecast.
In the following, we analyze the three named parameters for all forecasts of the winters 2000/2001-2019/2020 initialized
during the GL or ScTr regime.

### 4.1.2 Number of regime transitions during the forecasts

Regime changes might be more difficult to forecast than persistence. Therefore, we compare the number of regime changes
depending on the WR at initialization and the target date. The fraction of forecasts reaching a specific WR at the target date is
different depending on the WR at initialization. Thus, for every set of forecasts, we weigh the median regime changes per WR
present at the target date by the fraction of forecasts reaching the respective WR.

The weighted median of regime changes during the lead time of 14 days is 1.21 for forecasts initialized during the GL regime
and 1.22 for forecasts initialized during the ScTr regime (Fig. 3). This means that for forecasts initialized during the GL or ScTr
regime, on average, one or two regime transitions occur during the 14-day forecast period. Both regimes show a maximum of
four regime changes during the forecast. The interquartile range between the number of regime changes and the number of
outliers differ between the regimes present at the target date for forecasts initialized during the GL and ScTr regime. However,
overall the differences are small and therefore this aspect is unlikely an important explanation for the skill differences.



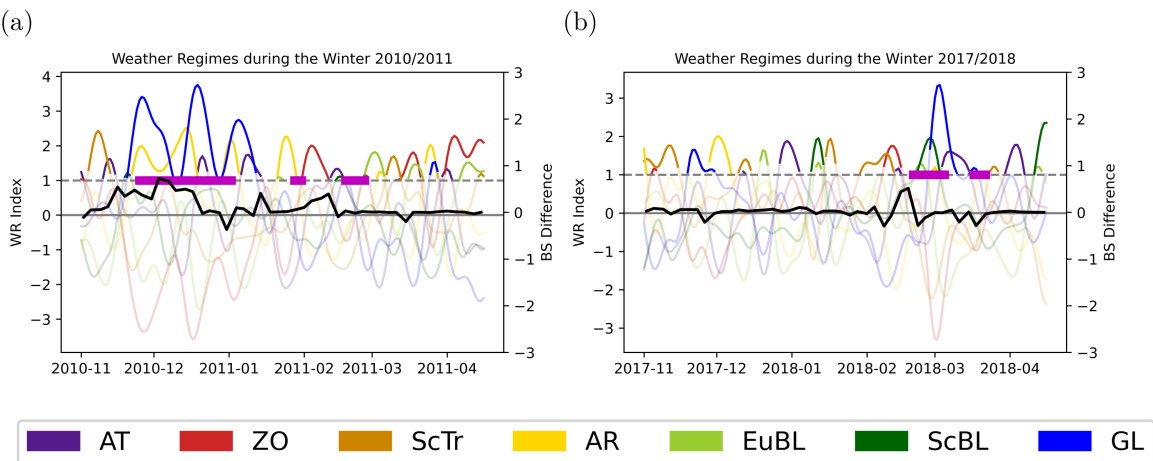

**Figure 2. WR index during two winters.** The WR Index is shown with a lag of 14 days (= WR at initialization) for the winters 2010/2011 (a) and 2017/2018 (b). The thick black line shows the difference in BS values of the mean-bias-corrected ECMWF's S2S reforecasts and the climatological benchmark ensemble. Positive values denote a better performance of the reforecasts. The magenta dashed line shows the occurrence of cold-wave days.

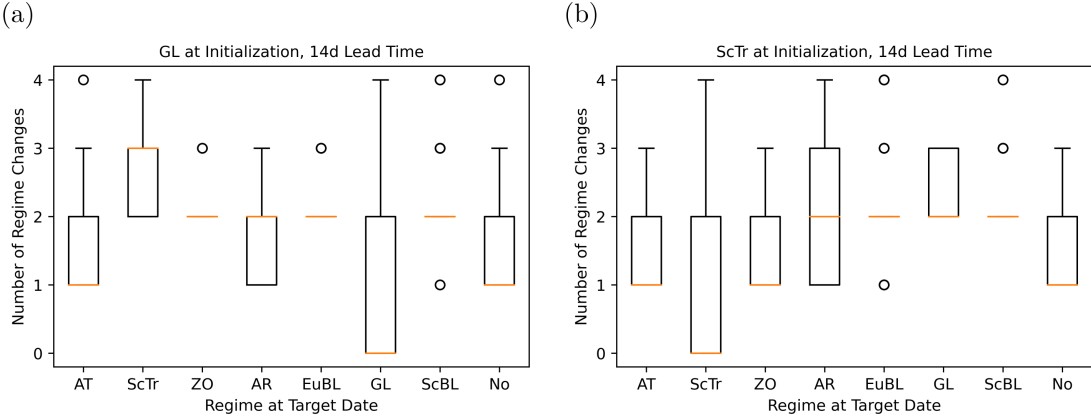

**Figure 3. Number of regime transitions during the forecast.** The number of regime changes are calculated for all forecasts of the winters 2000/2001-2019/2020 initialized during the GL (a) and ScTr regime (b) for a lead time of 14 days. The orange line marks the median of regime changes, the boxes the interquartile range and the open circles the 1.5 times interquartile range. Note: In order to increase the sample size, we use in the following all possible initialization dates of the winters 2000/2001-2019/2020 instead of only the dates where ECMWF's S2S reforecasts are initialized. This leads to a mixture of "hypothetical" forecasts (since no reforecasts are initialized at that date) and "real" forecasts.





### 4.1.3 Overall number of active WRs per day of lead time

As a next step, we investigate the WR successions during the forecasts. Here, we do this in a cumulative way focusing on the overall number of active WRs per day of lead time summed over all forecasts. The single actual WR successions are not considered. As an example, if on one day of the lead time either the AT or AR regime is present in all considered forecasts, the overall number of active WRs at this day of the lead time is two.

We hypothesize a higher predictability when the overall number of active WRs per day of lead time is low. Furthermore, we hypothesize a higher predictability if the fraction of a small number of WRs (e.g. two) is high (e.g. occuring in more than 50% of the forecasts) in comparison to the fraction of the other WRs. These numbers are determined for the forecasts of the winters 2000/2001-2019/2020, sorted by the WR present at initialization (either GL or ScTr) and target date (all WRs).

In 31.6% of forecasts initialized during the GL regime, the "No" regime is reached at the target date (Fig. 4 (e)). Thereby, at each day during the forecasts, in more than 60% of the cases either the GL or "No" regime is present. The second most often occurring regime at the target date is the GL regime itself (Fig. 4 (d)). Here, in more than 60% of the cases the GL regime is present at each day of the forecasts. Together, these two account for roughly half of all forecasts initialized during the GL regime. In another roughly 30% of all forecasts, namely the ones with the AR, AT and ZO regime at the target date, mainly two regimes occur during the forecasts but with varying fractions (Fig. 4 (a), (b) and (h)). The remaining cases show a more diverse picture with up to seven regimes per forecast day (Fig. 4 (c), (f) and (g)).

Likewise, for forecasts initialized during the ScTr regime, the "No" regime occurs most often at the target date (Fig. 5). Here, in more than 2/3 of the cases, either the ScTr or the "No" regime is present at each day of the forecasts (Fig. 5 (e)). The second most often occurring regime at the target date is the ScTr regime with the ScTr regime also present at each day of the lead time in a little less than 50% of the cases (Fig. 5 (g)) and the third most often occurring regime at the target date is the ZO regime with the ScTr or ZO regime present at each day during the forecasts in 70% of the cases (Fig. 5 (h)). The latter two regimes occur roughly equally often and make up approximately 60% of all forecasts, together with the ones with the "No" regime at the target date. During the other regimes at the target date, the fraction of the regimes during each day of the forecasts is more diverse (Fig. 5 (a), (b), (c) (d) and (f)).

Although a lower overall number of active WRs or respectively a high fraction of occurrence for a specific WR during the forecasts may increase skill in certain situations, we cannot find a general dependence of the forecast skill on the overall number of active WRs per day during the forecast.

### 4.1.4 Actual WR successions during the forecasts

Besides the overall number of active WRs at each day during the forecast, we investigate the role of the actual WR successions. This means, that we are now focusing on the single WR successions (e.g. "GL → AT") of the forecasts. Possibly, some WR successions are easier to forecast than others. We assume, that this is especially the case when WR succession follow typical climatological pattern. To test this, we firstly analyze all WR successions occurring during the winters 2000/2001 - 2019/2020. Then, we split the regime sequence during a forecast into parts of two WRs each (e.g. "GL → AT → No" is split into "GL





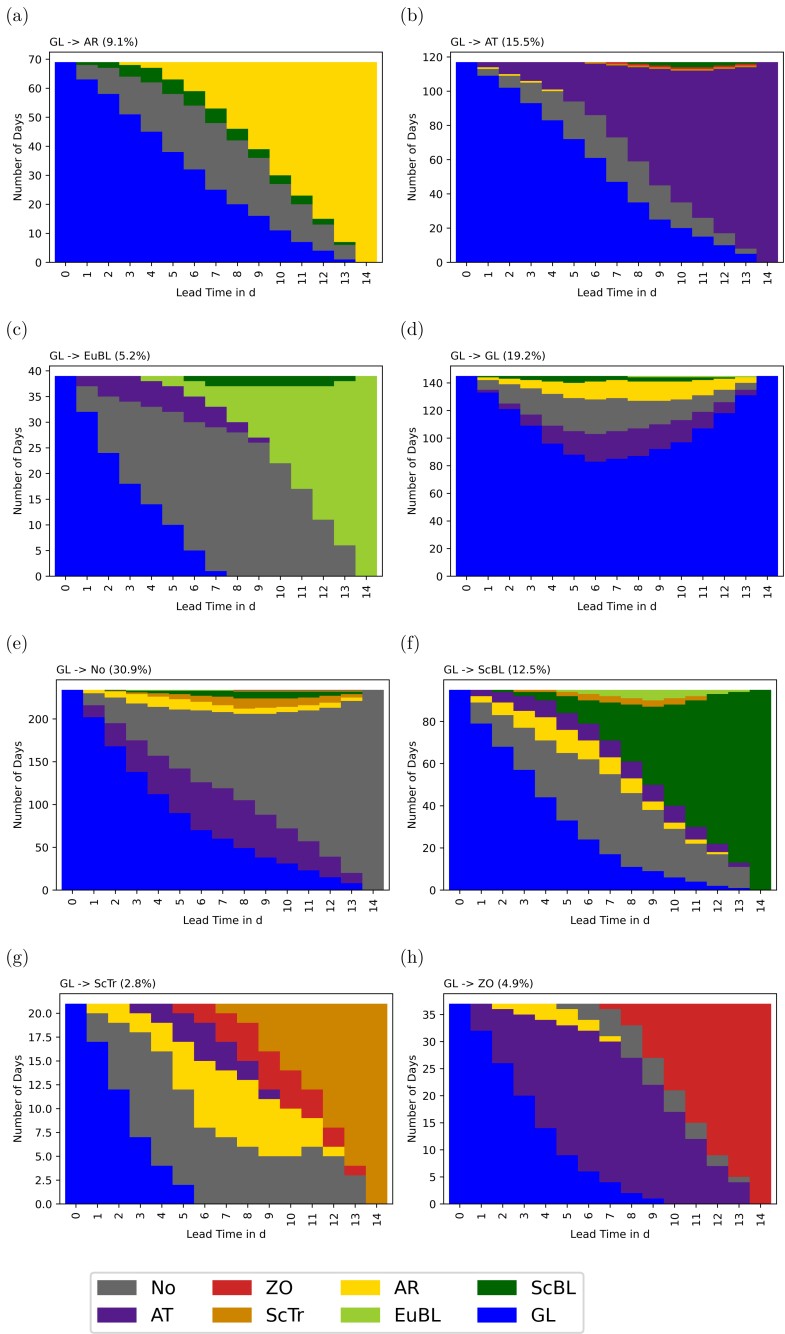

**Figure 4. Frequency of WRs for forecasts initialized during the GL regime with a lead time of 14 days.** The frequencies of WRs are shown for forecasts with the AR (a), AT (b), EuBL (c), GL (d), No (e), ScBL (f), ScTr (g) and ZO (h) regime present at the target date of the forecast. The fraction of forecasts reaching the different WRs at their target date are given in the titles above the subplots. Note: In order to increase the sample size, we use in the following all possible initialization dates of the winters 2000/2001-2019/2020 instead of only the dates where ECMWF's S2S reforecasts are initialized. This leads to a mixture of "hypothetical" forecasts (since no reforecasts are initialized at that date) and "real" forecasts.




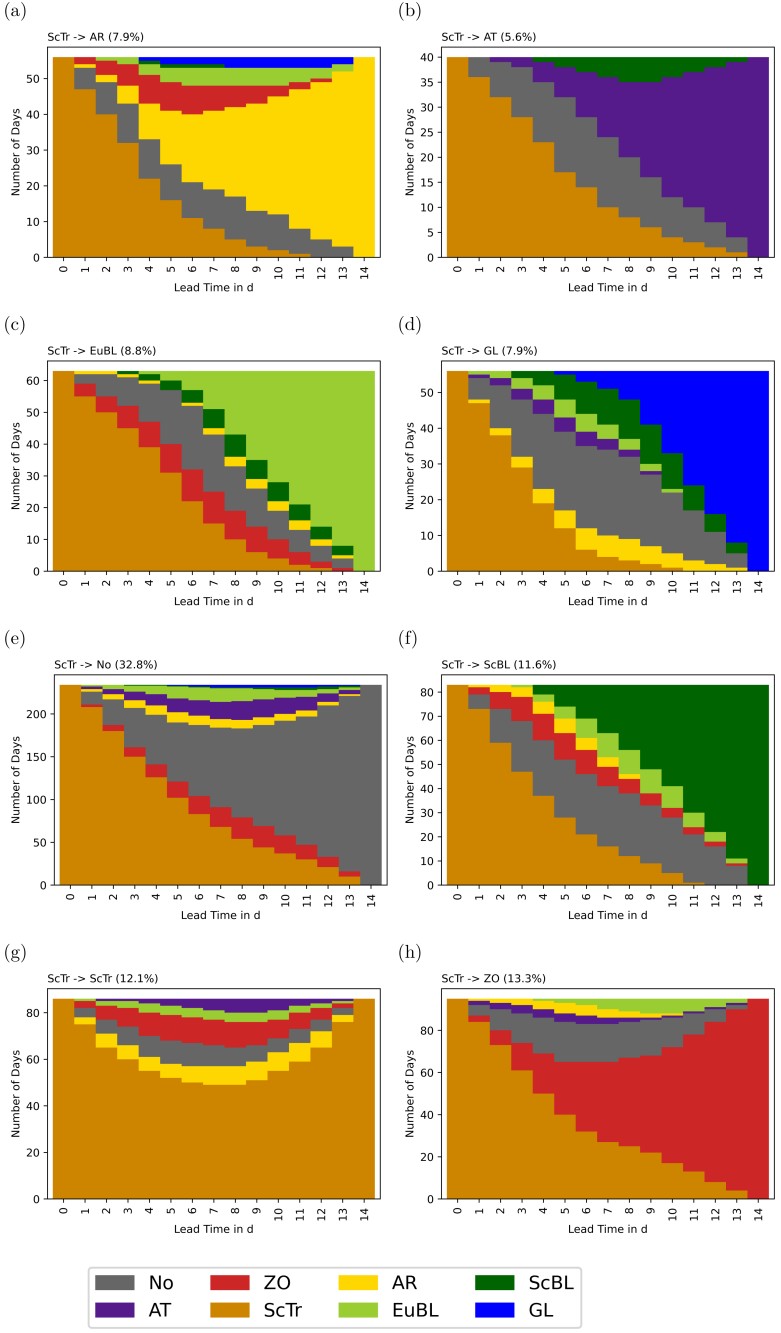

**Figure 5. Frequency of WRs for forecasts initialized during the ScTr regime with a lead time of 14 days.** The frequencies of WRs are shown for forecasts with the AR (a), AT (b), EuBL (c), GL (d), No (e), ScBL (f), ScTr (g) and ZO (h) regime present at the target date of the forecast. The fraction of forecasts reaching the different WRs at their target date are given in the titles above the subplots. Note: In order to increase the sample size, we use in the following all possible initialization dates of the winters 2000/2001-2019/2020 instead of only the dates where ECMWF's S2S reforecasts are initialized. This leads to a mixture of "hypothetical" forecasts (since no reforecasts are initialized at that date) and "real" forecasts.





→ AT" and "AT → No"). If all parts of the regime sequence during a forecast follow typical climatological pattern, the whole WR sequence is considered to do so. During the analysis, we focus on the sequence of regimes as a whole without taking persistence of the individual WRs per se into account. However, when only few WRs are present, the persistence of at least one WR must be high.

The most often occurring actual WR succession during the winters 2000/2001-2019/2020 is the transition from ScTr to the "No" regime (Fig. 6). It is followed by the successions "No → EuBL" and "AT → No".

     We define the top 11 successions present during the winters 2000/2001 - 2019/2020 as the "typical climatological patterns". We consider the top 11 instead of the top ten successions, because the tenth and 11th most often occurring WR succession have the same frequency.

For the forecasts initialized during the GL regime, the WR successions vary between three for ZO and EuBL at the target date and 13 for the "No" regime (Fig. 7). In case of the latter, two of these 13 options make up the majority of successions (Fig. 7 (e)). These are "GL → No" and "GL → AT → No", whereby the former is following typical climatological patterns. In case of the ZO regime at the target date, the most important is "GL → AT → ZO" which is not following typical climatological patterns (Fig. 7 (h)). When EuBL is present at the target date, the most often actual WR succession is "GL → No → EuBL"

which is following typical climatological pattern (Fig. 7 (c)). When the ScBL regime is present at the target date, the most often occurring WR succession is "GL → No → ScBL" (following typical climatological patterns), and when the AT regime is present at the target date is is "GL → AT" (Fig. 7 (b) and (f)). For forecasts with AR at the target date, the successions "GL → No → AR" and "GL → AR" are most often found and equally likely but only the former is following typical climatological pattern (Fig. 7 (a)). In case of the GL present at initialization and target date, it is persistence (Fig. 7 (d)) and in case of the

ScTr regime found at the target date, there is no clearly preferred succession.

     Persistence is also the most common for forecasts with the ScTr regime present at the target date when initialized during the ScTr regime (Fig. 8 (g)). Clear common actual WR successions are found for forecasts with the "No", ScBL and ZO regime present at the target date (Fig. 8 (e), (f) and (h)). These are "ScTr → No", "ScTr → No → ScBL" and "ScTr → ZO". With the exception of the latter, these actual WR sequences follow typical climatological patterns. A clear preferred WR succession is

not seen for forecasts with the AR, AT, EuBL and GL regime at the target date (Fig. 8 (a), (b), (c) and (d)).

     Considering only the most often actual WR successions per WR at the target date of the forecasts initialized during the GL and ScTr regime, we find that 61.6% of the forecasts initialized during the GL regime show actual WR successions that follow typical climatological patterns. Additionally, 18.1% of the forecasts show persistence. In case of forecasts initialized during the ScTr regime, 53.9% of the forecasts show actual WR successions following typical climatological patterns and 13.5% of

the forecasts persistence. This supports our hypothesize, that actual WR successions following typical climatological patterns might be easier to forecast and thus leading to an increased forecast skill. Furthermore, persistence of WRs seems to be a factor for an increase in skill.

     Overall, we have shown that neither the number of regime changes nor the overall number of active WRs at each day during the forecasts explains the differences in forecast skill of ECMWF's S2S reforecasts initialized during the GL and ScTr regime.





**Figure 6. Preferred actual WR successions during the winters 2000/2001 - 2019/2020.**

Instead, the actual WR successions during the forecasts play an important role. If the WR successions of the forecast follow
typical climatological pattern or persistence is found, the forecast skill is increased.







**Figure 7. Successions of WRs for forecasts of the winters 2000/2001-2019/2020 with a lead time of 14 days initialized during the GL regime.** The frequencies of WR successions are shown for forecasts with the AR (a), AT (b), EuBL (c), GL (d), No (e), ScBL (f), ScTr (g) and ZO (h) regime present at the target date of the forecast. Note: In order to increase the sample size, we use in the following all possible initialization dates of the winters 2000/2001-2019/2020 instead of only the dates where ECMWF's S2S reforecasts are initialized. This leads to a mixture of "hypothetical" forecasts (since no reforecasts are initialized at that date) and "real" forecasts.





**Figure 8. Successions of WRs for forecasts of the winters 2000/2001-2019/2020 with a lead time of 14 days initialized during the ScTr regime.** The frequencies of WR successions are shown for forecasts with the AR (a), AT (b), EuBL (c), GL (d), No (e), ScBL (f), ScTr (g) and ZO (h) regime present at the target date of the forecast. Note: In order to increase the sample size, we use in the following all possible initialization dates of the winters 2000/2001-2019/2020 instead of only the dates where ECMWF's S2S reforecasts are initialized. This leads to a mixture of "hypothetical" forecasts (since no reforecasts are initialized at that date) and "real" forecasts.





## 4.2 Differences in WR successions before the best (worst) predicted days within cold waves

In contrast to Subsect. 4.1, in this subsection only the "real" S2S reforecasts from ECWMF are considered since comparisons based on the BS of forecasts are made.

### 4.2.1 WR characteristic of all cold-wave days during the winters 2000/2001 - 2019/2020

Analogously as done for to the occurrence of (non-) cold-wave days, we investigate the WR characteristics during days within cold waves. The most frequent WR present at the start of a cold wave is the AR regime, which is found at roughly 30% of the cold wave starts during the winters 2000/2001-2019/2020 (Fig. 9 (b)). The GL regime is present at the start of approximately 1/5 of the cold waves, followed by the "No" regime, which is found in roughly 17% of the cases. All other regimes are less often present at the start of cold waves, whereby the EuBL and ScBL regime are found in roughly 11% of the cases each. The AT regime is not found at all at the start of the cold waves during the winters 2000/2001 - 2019/2020.

Up to four regime transition are observed during cold waves, whereby the median number of regime transitions is one (Fig. 9 (c)). The interquartile range is thereby between zero and two. Most often, a regime transition occurs in the second half of the cold wave after approximately 55% of the days (Fig. 9 (d)). Here, the interquartile range is between 35% of the days and the end of the cold wave.

Concerning the actual WR successions during cold waves, no clearly preferred successions are found (see Fig. S1 (a) in the supplementary material). Since air masses need a certain time to reach and influence Central European temperatures from their origin, the actual WR successions observed in the week before the cold wave starts are considered. However, this provides an even less clear picture (see Fig. S1 (b) in the supplementary material).

When looking at the WR index of the week before the cold waves instead of the categorical WRs, in the mean, the AR and GL regime show the highest values until one day before the cold wave start. However, the large spread of the WR indices shows that there is no clearly preferred WR (Fig. 9 (a)).

### 4.2.2 WR index of best and worst predicted cold-wave days

In order to analyze what distinguishes the best from the worst predicted cold-wave days, we use the continuous WR index instead of the categorical WRs. This is done, since a large proportion of the days is usually classified as "No" regime. Nevertheless, also during the "No" regime, the atmospheric conditions can vary substantially, which is depicted by the continuous WR index.

We compare the WR indices of the 10% best and worst predicted cold-wave days of the winters 2000/2001-2019/2020 independent of the WR present at their initialization. These are determined via the BS difference between ECMWF's S2S reforecasts and the climatological benchmark. In total, we analyze the 13 best predicted cold-wave days, of which 12 are intermediate days and one the last day of a cold wave. The days are scattered over eight different winter periods, whereby at most two days belong to the same winter and cold wave. Of the 13 worst predicted cold-wave days, eight are intermediate





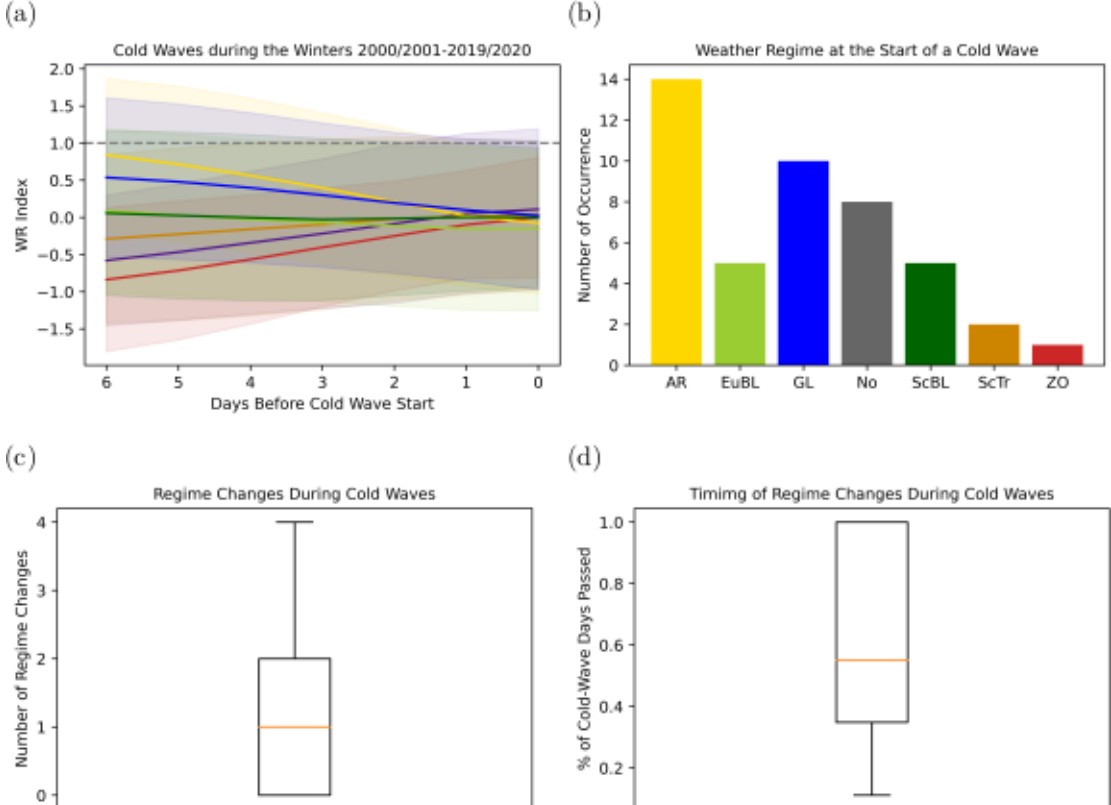

**Figure 9. WRs at the start of cold waves, number and timing of WR transitions during cold waves during the winters 2000/2001-2019/2020.** The median WR index (solid lines) and its spread (shading) is shown for the week before the cold wave start ((a), colors as in (b)). The categorical WR at the start of the cold wave is depicted on panel (b). Furthermore, the number of regime changes during a cold wave (c) and the timing of the regime changes (d) are shown.

days, three end and two start days of cold waves. These days are spread over nine different winters, whereby at most three days belong to the same winter and two to the same cold wave.

We concentrate our analysis on the WR index values above one, which indicate the presence of a well defined regime, and take the daily mean of those. In case of the best predicted cold-wave days, the AR regime is the most prominent regime one day before initialization until eight days after initialization (Fig. 10 (a)). The GL regime also has a high contribution during this time. Between six and three days before the cold wave starts, the EuBL regime is most prominent. In the remaining days before the cold wave, the ScTr regime is most pronounced.

For the 10% worst predicted cold-wave days, the GL regime is most prominent between 17 and 11 days before the cold-wave day occurs (Fig. 10 (b)). Then, the AR regime is the most pronounced WR until seven days prior to the onset of the cold wave, followed by the ScTr regime being most prominent until two days before the cold wave begins. In the last two days prior to the cold wave, both, the GL and AR regime are most pronounced.



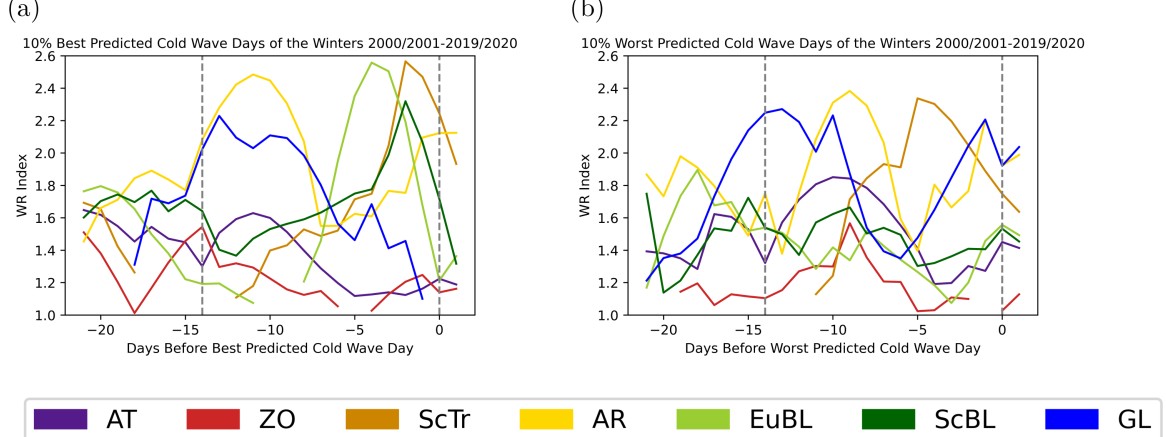

**Figure 10. Time series of the WR index of the best and worst predicted cold-wave days during the winters 2000/2001 - 2019/2020.**
The time series of the WR index of the 10% best predicted cold-wave days is shown on the left (a) and the time series of the WR index of
the 10% worst predicted cold-wave days on the right (b). Only the mean of values above 1 are shown. The vertical dashed lines show the
initialization of the forecasts and the respective cold-wave day.

According to a Welch's t-test, the difference of the AR and EuBL regimes between the best and worst predicted cold-wave
days is significant. Thus, we suggest that the occurrence of the EuBL regime in the week before the predicted cold-wave day
increases forecast skill. Furthermore, the timing of the occurrence of the AR regime seems to play a role.

### 4.2.3 Illustrative case studies of two cold waves

Given the considerable variability between individual cold waves in Central Europe, we select two case studies to show the
application of our results and to illustrate the characteristics of such events. These feature a best (worst) predicted cold-wave
day and the WRs present are close to the dominant WRs in the mean of the 10% best (worst) predicted cold-wave days.

One of the cold waves featuring a best predicted cold-wave day of the winters 2000/2001-2019/2020 is the one occurring
in January and February 2006 (Fig. 11 (a)). In this case, the ZO regime is present until two days before the initialization of
the forecast and then followed by the "No" regime, which persists until four days after initialization. During that time, the
strong zonal flow which leads to stormy and warm winter weather is continuously weakened. The former strong Icelandic Low
weakens, while simultaneously a strong high pressure system over the British Isles and southern Scandinavia evolves and the
EuBL regime is established. The longer the regime persists, the better the forecast skill gets.

The cold wave in January 2019 features one of the worst predicted cold-wave days of the considered 20 winters (Fig. 11
(a)). The atmospheric large-scale circulation in the week before the forecast initialization is almost equally close to both the
EuBL and AR regime. The strong high pressure system is located between Iceland and Scandinavia propagating to the north-
west while the AR regime is getting more pronounced in the seven days after the forecast initialization. At the same time, a
moderate Icelandic Low and a moderately strong high pressure system over the North Atlantic Ocean form during the ScTr





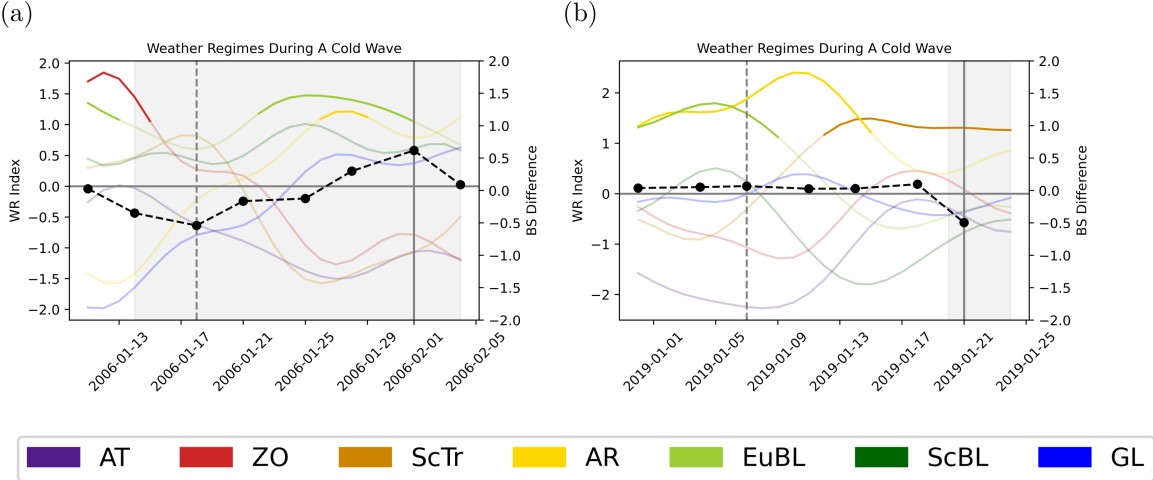

**Figure 11. WR index before and during two exemplary cold waves.** The WR index is shown for a cold wave with a one of the best predicted days in the 20-winter mean on the left (a) and for a cold wave with one of the worst predicted days in the 20-winter mean on the right (b). The black dashed line with the markers shows the BS difference of the forecast days. The vertical dashed show the initialization of the forecasts and the solid black line the best/worst predicted cold-wave day.

regime. Possibly, this is due to wave breaking and an easterly flow of air masses. This would explain the cold temperatures occurring in Europe although the atmospheric large-scale circulation is closest to the ScTr regime which is usually characterized by a zonally transport of warm airmasses across the Atlantic Ocean towards Central Europe.

### 4.2.4 Influence of the timing of a day inside a cold wave on the forecast skill


Since we use a temporal criterion in the definition of cold waves, which is that at least three consecutive days experience temperatures below a certain threshold, we assume that days in the middle or at the end of cold waves are generally better forecasted, since the atmospheric conditions leading to the fulfilment of this criterion are established longer.

To validate this assumption, we sort the days belonging to cold waves into three categories: the start days, the end days,
and the intermediate days. Depending on the length of the cold wave, the number of intermediate days varies. In total, we analyze 20 start days, 106 intermediate and 15 end days. The different number of start and end days results from the bi-weekly initialization of ECMWF's S2S reforecasts, which not necessarily falls together with the start or end of a cold wave with respect to the lead time of 14 days. We find that the mean-bias-corrected ECMWF's S2S reforecasts predict days during and at the end of a cold wave better than days at the beginning (Fig. 12 (a)). On average, intermediate and end days of cold waves
are predicted better in comparison to the climatological ensemble while days in the beginning of cold waves are not. However, these differences are not statistically significant.

In case of the RFC-based postprocessing model, only the intermediate days of cold waves are predicted better than by the climatological ensemble (Fig. 12 (b)). Their distribution of BS differences to the distribution belonging to the start days of




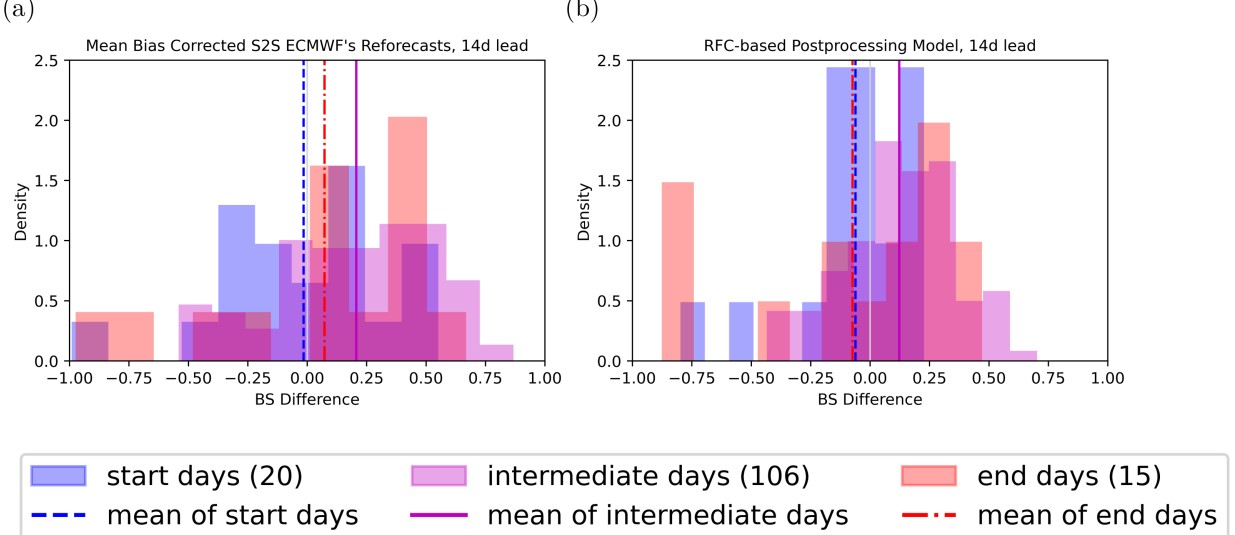

**Figure 12. BS differences of forecasts of cold-wave days.** The daily BS differences of the mean-bias-corrected ECMWF's S2S reforecasts (a) and the RFC-based postprocessing model (b) to the climatological benchmark ensemble are shown. The vertical grey line marks the BS difference at which both models predict equally well. A positive BS difference shows that the model performs better than the climatological benchmark ensemble.

the forecasts is statistically significant. However, the start and end days of cold waves are predicted less skillful than by the
climatological ensemble.

## 5  Summary and Conclusion

The objective of this study has been to analyze in how far WR successions during a forecast can be linked to subseasonal forecast skill and to explore the differences in WR successions before the best and worst predicted days within cold waves. In a previous study (Kiefer et al., 2024), we show that the binary forecasts of ECMWF's S2S reforecasts of the occurrence of
cold-wave days at a lead time of 14 days have a significantly better skill in the 20-winter mean when initialized during the GL regime compared to when the ScTr regime is present at initialization. These forecasts are further analyzed in the current study.

At the example of two illustrative winters which contain both, episodes of GL and ScTr, we derive three parameters based on the succession of WRs which can potentially be used to explain differences in forecast skill between forecasts initialized during GL and ScTr. These are the number of regime transitions during the forecast, the cumulative number of categorical WRs
at each day during the forecasts and the number of actual WR successions.

By investigating these parameters for the predictions of cold-wave days during the winters 2000/2001-2019/2020, we answer the first research question:

1. In how far can the WR succession during a forecast be linked to subseasonal forecast skill?





To increase the sample size for a higher robustness of results, we take all possible initialization days of the winters 2000/2001-
2019/2020 at which either the GL or ScTr regime are present into consideration and not only the dates on which ECMWF's S2S
reforecasts are initialized. This increases the possible number of forecasts by up to five per week. We find, that the number of
regime transitions during the forecast is not the main reason for an increased or decreased skill of the predictions. Although we
find differences in the number of regime changes between the WRs present at the forecast start, the weighted average median
of regime transitions is similar between forecasts initialized during the GL and ScTr regime. Furthermore, during both regimes
at initialization a similar amount of forecasts shows persistence in the WR which is hypothesized to be easier to forecast than
a large number of regime transitions.

The overall number of active WRs at each day during the forecasts is also not the main driver of an increase in forecasting
skill. Although we find large differences between the number of active WRs at each day of the forecast depending on the WR
at the target date, overall the number is similar between the two WRs present at initialization.

However, we find that a higher number of actual WR successions following typical climatological patterns leads to an in-
creased forecast skill. Although we cannot prove a causality here, this at least holds for predictions investigated in this study.
Hereby, we hypothesize, that WR successions occurring more often than others, which means that they follow typical clima-
tological patterns, are easier to forecast. Looking at WR successions has the advantage that besides the concrete succession of
WRs, also the persistence of WRs is represented by the number of different WRs during the forecast. Thereby, we concentrate
on the regime transitions as such and don't take the exact day of regime change into account.

In case of Central Europe in wintertime, arguably a correct forecast of cold-wave days is more important than the correct
forecast of a non-cold-wave day. Therefore, we also investigate the 45 cold waves observed during the winters 2000/2001-
2019/2020 separately. It is important to keep in mind that this is a rather small number of events, such that general statements
cannot be made. Due to the bi-weekly initialization of ECMWF's S2S reforecasts, only 20 start days, 15 end days and 106 in-
termediate days of the cold waves are included in the analysis. We find that intermediate days and days at the end of cold waves
are generally better forecasted than days at the beginning of a cold wave but the differences are not significant. Moreover,
there is no advantage in using an RFC-based postprocessing model in comparison to the mean-bias-corrected ECMWF's S2S
reforecasts.

Two thirds of the analyzed cold waves start when either the AR regime, the GL or the "No" regime is present at initialization.
The median number of regime transitions during the cold waves itself is one and happens in the second half of the cold wave.
Most often, the persistence of the AR regime or the transition from the GL to the "No" regime is found during a cold wave,
which follow typical climatological patterns. During the AR and GL regime, cold polar air masses are advected towards Central
Europe but during the "No" regime, warmer air masses are prevailing. It is important to kept in mind here, that the air masses
causing cold waves in Central Europe need a certain time to reach Europe from their origin, e.g. the Arctic. Therefore, we also
consider the WR successions in the week prior to the cold wave but a clear picture is not seen. Nevertheless, on a side note, we
find that the WR successions in the week before the cold waves start are following typical climatological patterns more often
when initialized during the GL regime than the ScTr regime. Additionally, the occurrence of a persistent GL regime during the




forecast is more often observed than a persisting ScTr regime. This might be another factor explaining the higher forecast skill of ECMWF's S2S reforecasts when initialized during the GL regime.

Independent of the WR at initialization, we compare the 10% best and worst predicted days within cold waves of the winters 2000/2001-2019/2020. Thereby, we focus on the following research question:

   2. What are the differences in the WR succession before the best (worst) predicted days within cold waves?

We find that the most important difference in the WR successions before the best predicted days within cold waves is the presence of the EuBL regime in the days before the target date instead of the ScTr regime which is present in case of the worst
predicted cold-wave days. The difference in the distributions of the WR index of the EuBL regime of the two forecast groups is significant. Interestingly, according to Osman et al. (2023), EuBL has a skill horizon of roughly 13 days and the presence of EuBL in case of the 10% best predicted cold wave days lies at the end of this time period. Besides the ScBL regime, which has the same forecast horizon, all other regimes can be forecasted skillfully on longer lead times.

At the example of two illustrative case studies we show, that the average results can also be found for individual cases. It has
to be kept in mind though, that Central European cold waves have a high variability in terms of WR successions as shown for the winters 2000/2001-2019/2020.

To further exploit the potential of WRs in enhancing subseasonal predictability, the influence of teleconnections on the WRs could be investigated in more detail. An example herefore is the study of Domeisen et al. (2020) who investigated the influence of SSWs on the probability of WR occurrence. During SSW events, stratospheric anomalies can propagate downward
into the lower stratosphere, influencing the upper-tropospheric circulation. This, in turn, allows a downward propagation of the disturbances finally reaching the mid-tropospheric circulation which is represented by the WRs. Further teleconnections include tropical disturbances such as arise from the El Niõ Southern Oscillation or the Madden-Julian-Oscillation. Additionally, the analysis of subseasonal forecasts of different NWP models might be beneficial to account for possible model biases. At last, the inclusion of further lead times on the subseasonal timescale will complete the picture.

*Code and data availability.* The used code for this study will be made available online when the paper is accepted for publication.

"This work is based on S2S data. S2S is a joint initiative of the World Weather Research Programme (WWRP) and the World Climate Research Programme (WCRP). The original S2S database is hosted at ECMWF as an extension of the TIGGE database".

"We acknowledge the E-OBS dataset (Cornes et al., 2018) from the EU-FP6 project UERRA (http://www.uerra.eu) and the Copernicus Climate Change Service, and the data providers in the ECA&D project (https://www.ecad.eu)." Special thanks Hendrik Feldman and the
"ClimXtreme" team for the courtesy of providing the E-OBS data used in this study. Christian Grams is gratefully acknowledged for providing the weather regime data.

*Author contributions.* The concept of the study was developed by SMK, PL and JGP. Data analysis and Figures were done by SMK. SMK wrote the original draft with help of JGP. All authors have contributed with methods, the discussion and revision of the original draft.



*Competing interests.* PK is an editor of WCD.

*Acknowledgements.* SMK was funded by the Transregional Collaborative Research Center SFB/TRR 165 "Waves to Weather" (www.wavestoweather.de) funded by the German Research Foundation (DFG). PL was funded by the German Helmholtz Association ("Changing Earth" program). SL gratefully acknowledges support by the Vector Stiftung through the Young Investigator Group "Artificial Intelligence for Probabilistic Weather Forecasting". JGP thanks the AXA Research Fund for support.



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
