# Peer review of "The Role of Weather Regimes for Subseasonal Forecast Skill of Cold-Wave Days in Central Europe"

_EGUsphere, 2024_

## Author Comment (AC1)

**Anwers to Comments from Referee 1**

Answers are written in green font.

This work described in this manuscript extends earlier studies by the same authors, focusing on a thorough analysis of the connection of weather regimes (and their succession) with the predictability of cold-wave days in Central Europe. The analysis shows that more common ('climatological') WR successions tend to be more predictable than uncommon WR successions, while other factors like the number of regime transitions between forecast initialization and valid time did not show a clear association with forecast skill. The paper is interesting, but some clarifications and more evidence for the main conclusion is required as detailed below.

**General comment:**

The main conclusion of the manuscript is that among the different WR-related explanations of increased/decreased predictability the frequency of WR successions following climatological patterns plays an important role. This conclusion is primarily based on the observation that 61.6% vs. 53.9% of a subselected set of cases follows such climatological patterns. That difference is noticeable but not huge, and given the additional complication due to the subselection criterion (only the most frequent WR successions per WR at the target date are considered), which presumably has the effect of amplifying the observed difference, I feel that more evidence for this conclusion should be provided. Would it be possible, for example, to calculate the Brier score for forecasts with the GL/ScTr WR at initialization time separately for the cases where the WR successions do and do not follow a climatological pattern and test whether the score differences are statistically significant?

Only a limited set of data is available for our analysis. Although the WR proposed by Grams et al. (2017) are calculated for the time period between 1979-2022, the calculations are only based on ERA5. There are no forecasts of the WR available to us. Furthermore, since ECWMF's S2S reforecasts of the 2-meter temperatures are only available bi-weekly and for the lasts 20 winters, this further constraints the sample sizes. Mixing reforecasts with different model versions is something we like to avoid since changes in the model set-up might dilute the results. Having said this, the resulting available sample sizes are in our eyes too small to allow for a sensible significance testing depending on the WR at the initialization and the valid date. Nevertheless, to get an impression, we grouped the reforecasts which follow typical climatological patterns and which not. The results are summarized in the table below. In total, they are 104 reforecasts initialized during GL and 129 reforecasts initialized during ScTr. Of the reforecasts initialized during GL, 54% follow typical climatological pattern. Of the reforecasts initialized during ScTr, 46% follow typical climatological pattern.

| GL at initialization | | | | ScTr at initialization | | | |
|---|---|---|---|---|---|---|---|
| WR at valid date | # follow. clim. patterns | # follow. not clim. patterns | % follow. clim. patterns | WR at valid date | # follow. clim. patterns | # follow. not clim. patterns | % follow. clim. patterns |
| AT | 4 | 12 | 25 | AT | 3 | 3 | 50 |
| ScTr | 1 | 2 | 33 | ScTr | 6 | 7 | 46 |
| ZO | 2 | 4 | 33 | ZO | 8 | 14 | 36 |

| AR | 6 | 11 | 35 | AR | 3 | 11 | 21 |
|---|---|---|---|---|---|---|---|
| EuBL | 2 | 2 | 50 | EuBL | 5 | 3 | 63 |
| GL | 20 | 11 | 65 | GL | 0 | 11 | 0 |
| ScBL | 7 | 1 | 88 | ScBL | 5 | 5 | 50 |
| No | 14 | 5 | 74 | No | 29 | 16 | 64 |

**Specific comments:**

- Section 2.2: Are the ECMWF reforecasts also temporally smoothed (like the observation data), or is that unnecessary due to the subsequent post-processing?

The reforecasts are not temporally smoothed for two reasons. First, we only use the forecast of the specific day of lead time and second, the subsequent postprocessing is already accounting for temporal uncertainties by taking the multi-year daily mean over 19 winters during the mean bias correction. Furthermore, due to the bi-weekly initialization of the reforecasts, the predictions we use are several days apart which makes the application of a running mean difficult. Additionally, for many socio-economic application the forecast of weather at a specific day is more important than a smoothed forecast, especially when it comes to extremes.

- 131: Aren't these just forecast errors of an ensemble mean forecast? I find it strange to call them biases, which to me is a systematic error, while without further aggregation the quantities calculated here contain (a substantial amount of) random forecast errors as well.

Yes, there are forecast errors but there is also a drift of the ensemble mean towards the model's climatology. This drift is in our understanding a bias and it is primarily this drift which we aim to remove from the predictions. Specifically, we substract the mean error (= systematic error = bias) of past forecasts compared to observations from the current predictions.

- Section 3.2, 2nd paragraph: More detail is required for this ERA5-based predictor. Is ERA5 data at the different hours from the day before initialization time used here? Can you briefly describe the preprocessing operations mentioned in 149?

The data at the different hours is retrieved at the day of initialization. We clarified this in the manuscript. I now states: "All meteorological fields are preprocessed by computing the minimum, mean, maximum and variance of each field before model training. This results in four predictors per meteorological variable at each time step. In case of ECMWF's S2S reforecasts, only the ensemble information (minimum, mean and maximum and their variances) instead of each individual ensemble members is taken into account. This leads to four predictors (instead of four predictors times eleven ensemble member) per meteorological variable at each time step. Furthermore, the minimum, mean, maximum and variance of the 2-meter temperature reforecast ensemble, averaged over Central Europe, is added as a predictor. The month is also added in order to account for the seasonality of temperatures and thus the occurrence of cold-wave days in winter."

- 189: I was very confused about this concept of 'hypothetical' forecasts when I read it here, and understood only later that it's not really a forecast, but that the weather regimes on these dates can still be analyzed. Maybe this can already be clarified here.

The paragraphs is changed to "In order to increase the sample size, we use in the following all days of the winters~2000/2001-2019/2020 instead of only the days where ECMWF's S2S reforecasts are initialized. This is done since the WRs on the days in-between initializations can still be analyzed. We refer to these as "hypothetical" forecasts (since no reforecasts are initialized at that date) and investigate them beside the "real" forecasts. For better reading, we use only the term "forecasts" in the following. We assume that the number of days on which each regime is present, 757~in case of GL and 713~in case of ScTr during winters~2000/2001-2019/2020, are similar enough to make a fair comparison. The analyzed WRs are based on ERA5 reanalysis data only."

- 235-236: I don't understand what is meant by 'single actual WR successions', and found this sentence very confusing. This paragraph is generally hard to follow, but it becomes clear what is studied here in connection with Figure 4. The aforementioned sentence, however, could easily be removed without loss of information.

We removed the unclear sentence from the manuscript and referenced Figure 4 in the beginning of the paragraph so the reader can directly refer to it.

- 267-268: I don't understand what is meant by 'without taking persistence of the individual WRs per se into account'. What if the WR at initialization time persists for the 14 days lead time? Please rephrase and/or explain.

We added "If one WR is persistent during all 14~days of the forecast, it is treated as a WR succession of only one WR for simplicity even if it is technically not a succession of WRs." to the manuscript.

- 275: Perhaps clearer to say '..., the number of possible WR successions varies …' √

**Typos and language:**

72: -> their skill √

130: Therefore -> To this end √

139: Either "the ECMWF S2S reforecast ensemble" or "ECMWF's S2S reforecast ensemble" √

159: Please check this reference, I have never seen a citation with a range of publication years before

The reference is the documentation of the python package I have used (https://skranger.readthedocs.io/en/stable/) and the time range the years of which the author had the copyright. I changed the reference to only the last year o the copyright to avoid confusion.

---

## Author Comment (AC2)

**Anwers to Comments from Referee 2**

Answers are written in green font.

This study attempts to relate Euro-Atlantic weather regimes (and their transitions) to the predictability of severe cold events in Central Europe in ECMWF forecasts. However, I have several major concerns with this study, which make it unsuitable for publication in its present form. In addition, I found the manuscript to be rather muddled and difficult to read.

**Main comments:**

(1) The authors claim to investigate how the transition of weather regimes "during a forecast" can be used to explain differences in the predictability of extreme cold events from forecasts initialised during different regimes. From my perspective, this should include some assessment of the links between cold events and weather regimes within the forecasts themselves. At the most basic level, it would have been useful to show how the frequency of cold events is modulated by regimes in both observations and forecasts. For example, a prerequisite for the presented analysis is a demonstration that forecasts accurately capture the location and magnitude of (lagged) relationships between the regimes and surface impacts. Instead, the authors limit their analysis to the observed regime behaviour during the forecast period. This leads to conclusions such as "WR successions following typical climatological patterns might be easier to forecast and thus leading to an increased forecast skill". However, this conclusion remains speculation while there is no assessment of the regime behaviour within the associated forecasts. In particular, the is no demonstration that the forecasts with increased skill for the prediction of cold events have actually simulated the observed regime transition. Although it is physically plausible to link large scale regimes to extreme cold events, there could be other explanations for the success of a given forecast that are unrelated to the regimes considered.

This is a very valid comment. We would like to have done an analysis as proposed but unfortunately forecasts of the seven WR by ECWMF are not available. Due to temporal and computational constraints we were not able to calculate the WR forecasts ourselves for all reforecasts of the extended winter seasons of the 20 winters we use in our study. Changing to the four classical WR provided by ECMWF's forecasts is also not possible since for these only forecast diagrams are available but not the data of the forecasted WR themselves.

(2) The introduction should include a clear and concise description of the hypotheses to be tested and the associated diagnostics. Some of this information is embedded in the results section and should be moved to the introduction to be structured something like: "We hypothesise that it is more difficult to forecast extreme cold events when Euro-Atlantic regimes exhibit the following characteristics within the forecast period: (i) XXX, (ii) XXX, (iii) XXX. We measure these characteristics using the following metrics...(i) XXX, (ii) XXX...". The authors also need to expand on their physical reasoning and explicitly discuss their interpretation of the links between regime transitions and forecast uncertainty. However, as stated in comment (1), some of these metrics should include information from the forecasts. It is not possible to conclude that cold event forecasts are good/bad because they have predicted a particular regime transition, without assessing this property of the forecasts.

Yes, we restructured the introduction as proposed. We also expanded on the physical reasoning but,

as stated in the answer to (1), can unfortunately not include information of WR forecasts since they are not provided by ECWMF.

(3) It is not clear why the random forecast calibration method is emphasised in the introduction and methods and then only mentioned in a single sentence of the results (line 377).

Yes, you are right, the results of the random forests only play a minor role in the context of the paper. However, we think it is interesting to show that the application of a machine learning model does not always lead to improvements in forecasts.

(4) The authors present a lot of descriptive statistics derived from very small samples (e.g. descriptions of figures 4 and 5). However, with eight classes (7 WR + Null regime) this means there are 64 possible transitions to consider and it is not clear if the derived "climatological" regime successions are statistically robust. For example, how do figures 4 and 5 compare when they are limited to the dates for which forecasts are available? The sampling uncertainty is something could be tested more easily using regimes derived from forecasts by subsetting from the available members, for example. Given the authors' conclude that "climatological successions" are easier to predict, I think they need to demonstrate that the climatology is robustly defined based on the available data.

Only a limited set of data is available for our analysis. Although the WR proposed by Grams et al. (2017) are calculated for the time period between 1979-2022, the calculations are only based on ERA5. There are no forecasts of the WR available to us. Furthermore, since ECWMF's S2S reforecasts of the 2-meter temperatures are only available bi-weekly and for the lasts 20 winters, this further constraints the sample sizes. Mixing reforecasts with different model versions is something we like to avoid since changes in the model set-up might dilute the results. Having said this, the resulting available sample sizes are in our eyes too small to allow for a sensible significance testing depending on the WR at the initialization and the valid date. Nevertheless, to get an impression, we grouped the reforecasts which follow typical climatological patterns and which not. The results are summarized in the table below. In total, they are 104 reforecasts initialized during GL and 129 reforecasts initialized during ScTr. Of the reforecasts initialized during GL, 54% follow typical climatological pattern. Of the reforecasts initialized during ScTr, 46% follow typical climatological pattern.

| GL at initialization | | | | ScTr at initialization | | | |
|---|---|---|---|---|---|---|---|
| WR at valid date | # follow. clim. patterns | # follow. not clim. patterns | % follow. clim. patterns | WR at valid date | # follow. clim. patterns | # follow. not clim. patterns | % follow. clim. patterns |
| AT | 4 | 12 | 25 | AT | 3 | 3 | 50 |
| ScTr | 1 | 2 | 33 | ScTr | 6 | 7 | 46 |
| ZO | 2 | 4 | 33 | ZO | 8 | 14 | 36 |
| AR | 6 | 11 | 35 | AR | 3 | 11 | 21 |
| EuBL | 2 | 2 | 50 | EuBL | 5 | 3 | 63 |
| GL | 20 | 11 | 65 | GL | 0 | 11 | 0 |
| ScBL | 7 | 1 | 88 | ScBL | 5 | 5 | 50 |
| No | 14 | 5 | 74 | No | 29 | 16 | 64 |

**Other comments:**

Title & throughout: The authors emphasise "subseasonal" forecasts, but the analysis is limited to a lead time of 14 days. I would call this "medium-range".

According to White et al. (2017), "subseasonal" starts at a lead time of 10 days so the lead time of 14 days is within "subseasonal" (Fig. 1, White, C. J. et al.: Potential applications of subseasonal-to-seasonal (S2S) predictions, Meteorological Applications, 24, 315–325, https://doi.org/10.1002/met.1654, 2017.).

Abstract: "These results can be used to assess the reliability of cold-wave day predictions" - this has not been demonstrated.

We demonstrate this by showing that forecasts initialized during GL are more skillful and thus more reliable than forecasts initialized during ScTr. Furthermore, also the result that the forecasting skill of a cold-wave day is improved when EuBL is present in the days before can be used to to assess the reliability of forecasts. If EuBL is present in the days before a cold-wave day, a forecast predicting a cold-wave day is more reliable than when another regime is present.

Abstract and throughout: "we investigate in how far the succession of WRs during a forecast can be used to explain skill differences of forecasts initialized during different WRs" - I assume this refers to skill differences in the predictability of extreme cold events. However, this is ambiguous and I initially assumed it referred to the predictability of regimes themselves. This happens elsewhere in the paper where it would improve clarity to refer to the "skill of cold extreme predictions" rather than just "skill".

The skill differences refer to the predictability of whether there is a cold-wave day occurring or not. We changed to a more accurate wording.

Line 24: "on that timescale" - ambiguous and unnecessary. Could delete "on that timescale, which comprises". √

Line 48: The discussion of Greenland Blocking can be linked negative NAO.

We did.

Line 59 (and elsewhere): I find it confusing to have results from previous studies by the same author described in the present tense. In some cases this is done in the results section, which makes it difficult to distinguish what is new in the present study.

The idea was to stick to the same tense during the whole manuscript. However, to improve clarity,

we changed to past tense for the description of the results of previous studies.

Line 65-66: At this point in the introduction it would be very useful to clearly describe the hypotheses to be tested and the associated diagnostics. See main comment (2). √

Line 67 (and elsewhere): given the focus on conditional statistics and specific regime transitions, which dramatically reduces the available sample size, why limit analysis to specific winters?

Since ECMWF's S2S reforecasts are produced on the fly, every set of reforecasts is (theoretically) calculated by a different model version. To be consistent within the reforecasts, we only take reforecasts from one cycle which consist of the 20 winters we limit our analysis to.

Line 75: What is the motivation for using RF calibration? Also see main comment (3). √

Section 2.1: This would benefit from separation into two sections containing (i) the description of the index to be predicted (i.e. methodology that is common to both forecasts and observations) and (ii) the construction of climatological ensemble.

There are two "methods" we use for constructing the climatological benchmark ensemble and the forecasts. The first one is simply taking the 2-meter temperature averaged over Central Europe, the second one the calculation of cold-wave days. If we put this information into a separate subsection, the subsection describing the climatological ensemble would only consist of "the climatological ensemble comprises the 2-meter temperature timeseries of the winters 1970/1971-2019/2020 in case of the continuous predictions and of the occurrence of cold-wave days in this time-period in case of the binary predictions." In our opinion, this is too little for a separate subsection so we would like to keep this subsection as it is.

Figure 1: The choice of colour scale means there is no distinction between ocean and land > 800m. Perhaps highlight the cold wave index region with a red contour or similar?

Yes, it does and it is also on purpose. The region used for averaging are only the coloured grid points. Mountains > 800m and the first coastal grid points are excluded. Ocean grid points are not provided by the E-OBS dataset.

Line 96: Which IFS cycle is use?

ECMWF's S2S reforecasts are computed by the model version CY46R1. We added this information.

Line 97: Are forecast data processed identically to the climatological ensemble? Is the same 7-day smoothing applied? If forecasts and reference data are not processed identically, this will systematically impact differences in Brier Score.

The reforecasts are not temporally smoothed since we only use the forecast of the specific day of lead time. Due to the bi-weekly initialization of the reforecasts, the predictions we use are several days apart which makes the application of a running mean difficult. It has been a trade-off between saving computational resources and downloading multiple lead times for a temporal smoothing. Furthermore, for many socio-economic application the forecast of weather at a specific day is more important than a smoothed forecast, especially when it comes to extremes.

Section 2.3: This section could be shorter if it is based on the exact same methodology as Grams et al.

It is not based exactly on Grams et al. (2017) since another dataset (ERA5) is used to calculate the WR.

Section 3.1 This could be part of the description of the IFS data.

Yes, you are right. However, since we want to emphasize that we use the mean bias correction as a postprocessing step instead of using the original reforecasts, we would like to keep this subsection separately.

Line 130: "Reforecast ensemble [mean]"?

Yes, the daily mean of the reforecast ensemble which is in our opinion equivalent to the "reforecast ensemble mean".

Line 131: I would rephrase this to "time series of errors" and use "bias" to describe the average/expectation of the error over many cases.

We changed accordingly.

Section 3.2: I find it extremely odd to have a long section dedicated to the description of a calibration method that is then only mentioned in passing in the results section. See also main comment (3). The authors should either (i) provide a clear motivation of the use of the calibration method and how it helps understand the link between regimes and cold wave predictions and adequately discuss the results or (ii) remove this element of the paper.

Please see response to (3).

Line 165: This notation of the Brier score is quite arcane. It is typically presented as the expectation of squared differences in forecast/observed probability for a specific event.
https://en.wikipedia.org/wiki/Brier_score

We explain the chosen representation in the text. However, if you think it is necessary to change to another notation, we could do that.

Line 173: What is the motivation for use of BS differences rather than the more typical skill score form i.e. BSS = (BS_benchmark – BS_model)/BS_benchmark)?

Skill scores are mean values and have therefore no meaningful values on a daily basis. Due to this, we use BS differences and not the BSS.

Line 180: The results section begins with a statement of results from a previous study by the same author in the present tense. It is then extremely ambiguous as to whether the following sentence ("We find that the ..." is another result from the previous study or something new. It would be easier for the reader if previous results are kept in the past tense and initially described in the introduction, with reference as required in the results.

We changed to the past tense for describing results of the previous study.

Lines 188-190 (and elsewhere, including figure captions): The description of observational data as "hypothetical forecasts" is extremely confusing. For example, Figure 3 is titled "Number of regime transitions during the forecast", which is confusing for two reasons: (1) this is not forecast data and (2) it is not limited to forecast dates. A more accurate description would be "Number of regime transitions in ERA5 within a 14 day moving window". Similarly, Figure 4 is title "Frequency of WRs in forecasts initialised during the GL regime". Again, this is not an accurate description. For this description (and associated "lead time" axis labels) to make sense, the authors would need to show the regime frequencies from the forecast model. It would then also be useful to compare with the observations limited to the dates of forecast initialisation.

We know that it is a complex wording but we did not come up with a better one. We thank you for your suggestion, however, it is not accurate in our opinion to use the expression "in ERA5 within a 14 day moving window " since we are only considering start days where either GL or ScTr is present. This description leads to the impression that every day is considered as a start day which is not true. Please, see furthermore the responses to (1) and (4).

Line 195: Why GL and ScTr only?

Because skill differences between the reforecasts initialized during those two regimes are significant.

Line 197: "GL is the dominant regime at the initialisation for forecast predicting the occurrence of (non-)cold-wave days" - what does this mean? GL is dominate regime for both cold-wave and non-cold-wave days?

This means, that for forecasts predicting the the mentioned time (mid-November to mid-January) of the winter, the GL regime is active at their initialization. So the GL regime is active 14 days before a prediction of a day in mid-November is done. This is true for both, days classified as cold-wave days and days classified as non-cold-wave days.

Line 209: "GL regimes tend to be more persistent" - is this result robust for the limited sample size?

Due to the limited sample sizes we don't think that the testing of significances makes sense. See also the answer to (4).

Lines 203-220 & 235-240: These paragraphs contain hypotheses that would be useful to include in the introduction, alongside additional discussion of how this links to forecast/skill uncertainty and description of the the proposed diagnostics.

We added this to the introduction.

Line 212: "number of active WRs regimes at each day" - what is the physical interpretation of multiple active regimes per day? If there is no clear distinction is there really a regime?

This is an accumulated statistic over many days. The accumulation is done for all "lead time 1" days , all "lead time 2" days, etc..

Line 223: "Regime changes might be more difficult to forecast than persistence" - again, this is speculation/hypothesis that should be introduced earlier rather than part of the results.

We added this to the introduction.

Figure 3: Plotting the median and interquartile range seems unnecessary for the small sample sizes. It would be more transparent to show every data point. For example, what does it mean in the case of GL->Zo and GL->GL that there is no interquartile range box? I assume there it means all contributing data points are equal but it would be clearer to just show the data.

Yes, it means that all contributing data points are equal to the median besides one outlier. However, showing only the data would lead to crowded plots in case of some of the transitions (there are up to 45 forecasts per category) so we opted to show boxplots instead.

Line 306: "Analogously as done for to the occurrence of (non-)cold-wave days, we investigate the WR characteristics during days within cold waves." I don't understand the distinction between (non-)cold-wave days and days without cold waves. Surely the set of non-cold-wave and cold-wave days includes all days?

Yes, (non-)cold-wave days is used to denote the set of non-cold-wave and cold wave days. Days within cold waves are the cold-wave days. Here, we only consider those.

Summary and conclusions: I don't think the authors can answer the proposed research question how is the "WR succession during a forecast linked to [...] forecast skill" without looking at weather regimes in forecasts.

We are aware that we only consider a limited sample size and only observations. However, as stated in the answer to (1), WR forecasts of the seven WR are not provided by ECMWF.

Lines 443-449: The links between weather regimes and teleconnections are interesting but this topic should have been introduced earlier (e.g. in the introduction). I agree it may be interesting to investigate the modulation of weather regime transitions/successions by other modes of variability, but the sample sizes is a major challenge. The authors already implicitly stratify data into 64 possible cases (GL-Zo ...etc), which leaves very little data for further stratification by MJO/polar vortex phase, for example. Maybe the authors could consider what would be required to make progress in this area (e.g. reforecasts with very many start dates covering long periods?)

Yes, we elaborated more on the outlook. The teleconnections are mentioned in the introduction where it is written that certain WR follow often in the weeks after an SSW event. However, we can include more links if needed.